# Methods of Analysis and Identification of Betulin and Its Derivatives

**DOI:** 10.3390/molecules28165946

**Published:** 2023-08-08

**Authors:** Altynaray T. Takibayeva, Gulistan K. Zhumabayeva, Abdigali A. Bakibaev, Olga V. Demets, Maria V. Lyapunova, Elena A. Mamaeva, Rakhmetulla Sh. Yerkassov, Rymchan Z. Kassenov, Marat K. Ibrayev

**Affiliations:** 1Department of Chemistry and Chemical Technologies, NJSC Karaganda Technical University Named after Abylkas Saginov, Karaganda 100027, Kazakhstan; altynarai81@mail.ru; 2Faculty of Natural Sciences, L.N. Gumilyov Eurasian National University, Astana 010000, Kazakhstan; gulistan2009@mail.ru (G.K.Z.); erkass@mail.ru (R.S.Y.); 3Chemical Faculty, National Research Tomsk State University, 634028 Tomsk, Russia; bakibaev@mail.ru (A.A.B.); lyapunova.mari@mail.ru (M.V.L.); 4Chemical Faculty, National Research Tomsk Polytechnic University, 634050 Tomsk, Russia; mamaeva.elena@mail.ru; 5Department of Organic Chemistry and Polymers, Chemistry Faculty, NJSC Karaganda University Named after Y.A. Buketov, Karaganda 100024, Kazakhstan; r_z_kasenov@mail.ru (R.Z.K.); mkibr@mail.ru (M.K.I.)

**Keywords:** betulin, the solubility of pentacyclic triterpenoids, bioavailability, chemical transformation

## Abstract

This scientific work presents practical and theoretical material on the methods of analysis and identification of betulin and its key derivatives. The properties of betulin and its derivatives, which are determined by the structural features of this class of compounds and their tendency to form dimers, polymorphism and isomerization, are considered. This article outlines ways to improve not only the bioavailability but also the solubility of triterpenoids, as well as any hydrophobic drug substances, through chemical transformations by introducing various functional groups, such as carboxyl, hydroxyl, amino, phosphate/phosphonate and carbonyl. The authors of this article summarized the physicochemical characteristics of betulin and its compounds, systematized the literature data on IR and NMR spectroscopy and gave the melting temperatures of key acids and aldehydes based on betulin.

## 1. Introduction

Biologically active compounds based on plants and products of their chemical transformation are used as food additives to maintain and promote a healthy lifestyle, prevent diseases and treat ailments. The nutritional value and pharmacological effect of such drugs primarily depend on their chemical composition, which is complex and multicomponent in nature, which is accompanied by difficulties in isolating, analyzing and identifying the main components and in their rationing.

Among the many extractive substances from plant raw materials, pentacyclic triterpenoids from birch bark, especially betulin (**1**) and related compounds (**2**–**9**), occupy a special place. Betulin (**1**) and betulinic acid (**5**), as a result of a number of chemical transformations, lead to highly active derivatives, some of which are comparable to clinically used preparations [1,2], and the content of betulin overwhelmingly predominates the content of betulinic acid in the native extract [3]. Moreover, betulin and betulinic acid are valuable matrices for many semisynthetic derivatives that are potentially more effective drugs [2,4,5,6,7,8]. Betulin exhibits many other biological effects (anti-inflammatory [9,10], antiviral (including anti-HIV), etc.) [6,11,12,13,14,15,16,17,18], including the antimycotic and antimicrobial activities mentioned above. Betulin also inhibits cholesterol and fatty acid biosynthesis and thus reduces diet-induced obesity. This compound reduces the size and improves the stability of atherosclerotic plaques (as evidenced by the reduction in macrophage accumulation) [19]. The authors of [19] also recommend its use in the treatment of type II diabetes by activating the sensitivity of cells to insulin. Pentacyclic triterpenoids can be used and developed as new drugs with a wide clinical effect [4,6]. There have been new reports [20,21,22,23,24] on the biological properties of betulin and its derivatives in recent years.

Thus, high-purity betulin (**1**) is widely used for the production of pharmaceutical and cosmetic products and biologically active additives. Betulin and its semisynthetic derivatives have a very high potential for use mainly in medicine [16,18].

Betulin (**1**) can actually be considered the first biologically active substance extracted from a plant source on an industrially significant scale [25]. Currently, it is considered that the outer bark of birch (birch bark) is rich in such pentacyclic triterpene compounds such as betulin (lup-20(29)-en-3β,28-diol), lupeol (lup-20(29)-en-3-ol), betulinic acid (3β-hydroxy-20(19)-lupaene-28 acid) and other minor components (oleanolic acid, ursolic acid and betulin aldehydes) [18]. It is generally assumed that birch bark is the area of bark exiting from the last formed layer of the periderm to the outermost surface of the tree [26].

Betulin (**1**) is found in warty birch or hanging birch (*Betula verrucosa* Ehrh. or *Betula pendula* Roth.) and fluffy birch (*Betula pubescens* Ehrh.), the most widespread in Russia and in the Northern hemisphere [27]. Betulin (**1**) is also found in the birch bark of white birches (*Betula alba*) growing in Europe [28]. The main source of betulin is birch bark [29,30,31], although betulin (**1**) has been found in at least two dozen plants belonging to various genera and families. For example, betulin (**1**) is found in hazel bark, calendula and other medicinal plants. For example, betulin, lupeol and betulinic acid are extracted from the bark of alder cordial (*Alnussubcordata* L.) [32], unabi vulgaris (*Ziziphus jujube* M.) [33,34] as well as from the ground part of thistle (*Atractylis carduus* L.) [35]. Betulinic acid is found in the leaves of plumeria (*Plumeria obtusa*) [36], triadenum (*Triadenum japonicum*) [37], orchid (*Orchid lusiaindivisa*) [38] and other plants (*Dillenia papuana*, *Tryphyllum peltatum*, *Ancistrocladus heuneaus*, *Diospyrus leucomelas*, *Tetracera boliviana*, *Sizyphus joazeuro*, *Syzigium claviflorum*, *Aerua javanica*) [39,40].

### 1.1. General Information about Betulin and Its Derivatives

Betulin is a natural pentacyclic triterpenoid belonging to the lupan group and has a systematic name–3β,28-dihydroxy-20(29)-lupen or lup-20(29)-en-3β,28-diol (Figure 1). 

The structure of the lupane series triterpenoids is based on a complex polycyclic system with a cyclopentaneperhydrophenanthrene (sterane) core consisting of four cyclohexane and one saturated cyclopentane ring condensed together. The rings have the designations **A**, **B**, **C**, **D** and **E** in accordance with the accepted numbering of the carbon skeleton atoms and the increasing numbers of carbon atoms (Figure 1). The presence of the five-membered ring **E** and an α-isopropenyl group in the C-19 carbon atom is characteristic of the betulin (**1**) belonging to this group [41].

Birch bark has two clearly distinguishable parts—outer and inner. The outer part of the bark is the richest in extractive substances. The main component of almost all extracts is betulin (**1**), which causes the white color of the bark. Lupeol (**2**) is a constant companion of betulin during its isolation (Figure 2). Birch bark extracts also contain, along with betulin (**1**), its oxidized derivatives: betulin aldehyde (**3**), betulonic aldehyde (**4**), betulinic acid (**5**), betulinic acid methyl ester (**6**) and 3-oxobetulinic (betulonic) acid (**7**) (Figure 2).

The chemical composition of the bark of many birch species, namely *Betula pendula* Roth. (*B. Verrucosa* Ehrh.), Betula pubescens Ehrh. (*B. alba* L.) and *B. davurica* Pall., has been studied in sufficient detail [42]. The content of betulin in the outer bark varies from 10 to 40% depending on the type of birch, the place and conditions of its growth, the age of the tree, the season and other factors [30,43,44]. The variety of properties of betulin is determined by the origin of this substance: plants synthesize betulin to protect against all adverse environmental factors and accumulate it exclusively in their shell. Betulin is a necessary part of plant shells to protect the plant from damaging environmental factors such as radiation, bacteria, fungi, viruses and insects. The reliability of the protective properties of birch bark is evidenced by the fact that it has been preserved without rotting in the soil for more than 1000 years, as evidenced by Novgorod birch bark certificates.

Betulin (**1**) is an odorless white crystalline powder with amorphous properties. The use of most solvents does not allow the growth of large crystals, which is a consequence of the presence of a large number of low-polar alkyl lipophilic fragments in the betulin molecule. At the same time, betulin (**1**) has one double bond and two hydroxyl groups in the molecule: primary and secondary. Both are associated with alkyl fragments and exhibit classical alcoholic properties. Therefore, betulin has a greater affinity for medium-polarity solvents than for low-polar solvents. The molecular weight of betulin (**1**) is 442.72 g/mol, the gross formula is C_30_H_50_O_2_ and the melting point is 261 °C, [α] +20 °C (in pyridine) [29,45], 258 °C (according to Gausman) [46]. This compound is resistant to oxygen and sunlight and is non-toxic. Betulin (**1**) does not occur in its free form [47]. Betulin (1) is quite inert—it does not dissolve with water; when isolated, it crystallizes into whitish prismatic crystals. It is relatively well soluble in boiling alcohols, ether, chloroform, benzene, pyridine, tetrahydrofuran and dimethyl sulfoxide [46]. The solubility of betulin decreases in alcohol solvents in the following series: 1-butanol > 1-propanol > ethanol > 1-pentanol > 1-hexanol > methanol. The solubility of betulin in esters decreases in the following decreasing order: ethyl acetate > methyl acetate > ethyl formate = methyl formate [48].

Betulin is characterized by its low solubility in nonpolar organic solvents [49,50]. Nevertheless, the range of solubility of betulin in various solvents should be noted; for example, the solubility of betulin in acetone (polar) is 5.2 g/L, and in cyclohexane (nonpolar), it is only 0.1 g/L at 15.2 °C, and at 35.2 °C, the solubility is 13.7 g/Land 0.67 g/L, respectively.

The comparison of the results [51] obtained for two types of solvents shows that nonpolar solvents have a number of advantages over their polar counterparts. Nonpolar solvents do not form solvates compared to polar solvents after recrystallization. The betulin molecules in polar solvents form hydrogen bonds with each other in addition to forming hydrogen bonds with solvent molecules [51].

Birch bark contains oleanane and ursan triterpenes along with lupan derivatives. Oleic acid and its derivatives predominate in some birch species, for example, in the black-crowned birch (*Betula dahurica* Pall.) [25,52].

The product of betulin rearrangement, allobetulin (**8**) (Figure 3), should be noted out of the other oleane derivatives found in birch bark extracts. Allobetulin (**8**) is easily obtained under the action of acidic agents [22].

The qualitative composition of the triterpenoids of the bark of black-backed birch *Betula pubescens* is the same as that of white-barked, but their quantity is 2–3 times less. At the same time, the content of betulin (**1**) in white- and black-barked *Betula pubescens* is significantly lower than in the bark of *Betula pendula* [53]. The main component of birch bark extract in all cases is betulin (**1**) (Figure 4).

Extracts of the inner bark of birch contain a small amount of betulin (**1**); however, they can be used as a source of phenolic compounds for certain purposes [27]. But, at the same time, the total extract of isolated triterpenoids from the inner bark of birch primarily includes betulin (**1**) (60–85% of the composition), as well as the currently well-studied lupeol, lupenone, uveol, betulin acetate, allobetulin, isobetulenol, oleanolic acid and others. The physicochemical properties of the above satellites of betulin are close to their progenitor (**1**).

### 1.2. Structure and Physico-Chemical Properties of Betulin and Its Derivatives

The properties of betulin (**1**) and its derivatives are largely determined by the structural features of this class of compounds and their tendency toward dimer formation, polymorphism and isomerization. Betulin (**1**) is a triterpene alcohol of the lupane series in which cyclohexane rings are in the “chair” conformation and the cyclopentane ring is in the “half-chair” conformation (Figure 5).

The spin-spin interaction constants observed in the ^1^H NMR spectra confirm the conformation of the “chair” of the six-membered betulin (**1**) rings. In addition, the geometry of the betulin molecule is determined by the location of hydrogen atoms and functional groups attached to the main nucleus: α- or β-orientation in space and α- and β-epimers at the C-3 position [54]. The hydroxyl group at C-3 in the equatorial position (β-OH) gives the betulin molecule greater thermodynamic stability than in the axial position (α-OH) since there is intensity in the α-position due to the repulsion of neighboring atoms.

The combination of these causes gives betulin (**1**) conformational lability (thermodynamic instability), which depends on the chemical and physical properties of the medium. Moreover, there are changes in the structure of not only the site in contact with another molecule but also the conformational state of betulin (**1**) as a whole. In addition, conformational lability determines the conformation of the “chair” of cyclohexane cycles, which are displaced relative to each other. Transannular interactions between unbound hydrogen atoms reduce the stability of the molecule, especially when the atoms are close to each other. Conformational lability also arises due to the mobile structure of the connection of cycles, the presence of four angular methyl groups, two hemdimethyl groups and terminal methylene group. In addition, betulin (**1**)-like related triterpenoids are sensitive to the effects of various factors causing the migration of methyl groups and protons between cycles. The discussed structural features of betulin (**1**) and its derivatives (**2**–**8**) cause not only high lability but also the theoretically possible existence of optical isomers and epimers as well as the ability to isomerize.

Intermolecular interactions of two thermodynamically unstable betulin (**1**) molecules cause various rearrangements and migrations of methyl or hydroxyl groups and also determine the most important physico-chemical properties of betulin (**1**) and its derivatives (**2**–**8**), including solubility. Intermolecular interactions, including both hydrophobic binding and the formation of H-complexes with organic solvents in triterpenoid molecules in the solid phase, lead to various polymorphic and solvatopolymorphic forms. Thus, the solvate of betulin (**1**) with ethanol in a composition of 1:1 has a rhombic syngony in the spatial group P212121, Z = 4. Betulin molecules are connected to each other and to ethanol molecules by hydrogen bonds; as a result of this, layers perpendicular to the crystallographic direction c are distinguished in the structure. All molecules are connected by hydrogen bonds inside the layer, and only Van der Waals interactions exist between the layers [51].

The most stable polymorphic form is the most energetically favorable under certain crystallization conditions, and the more ordered one usually has the worst solubility. Metastable solvatopolymorphs, although they have better solubility, can be toxic due to the presence of solvent molecules in them.

The formation of betulin dimers and inclusion complexes is probably explained by the desire to adopt an energetically favorable form characterized by a minimum of free energy due to hydrophobic interactions and Van der Waals forces of closely connected atoms (Figure 6).

The pseudopolymorphic forms (solvates) are isolated from solvents of different polarities during the crystallization of betulin (**1**), which, according to X-ray diffraction analysis, are inclusion complexes. Studies of the XRD patterns of betulin solvates and its derivatives with various solvents were carried out in [55,56,57,58,59,60,61]. Analysis of these data suggests that they have a clathrate-type structure. Since solvents enter the betulin crystal lattice and bind to betulin molecules in a certain way, the most stable are betulin (**1**) solvates with butanol, ethyl acetate, chloroform and dichloromethane. Betulin solvates recrystallized from acetone, methanol, ethanol and propanol are unstable and lose solvent when stored even at room temperature. In general, the crystallization of betulin and its derivatives is the most favorable of alcohol solvents with greater polarity because it does not change its structure. The crystal structure of betulin solvates is stabilized due to hydrogen bonds [53,62], and in this state, they can have independent pharmacological significance.

In general, the existence of unstable solvates and solvatopolymorphs or other polymorphic forms is a big problem in the pharmaceutical industry since such samples of polymorphs are usually able to change over time, sometimes completely disappearing and turning into another polymorphic form with lower solubility. 

Information on the solubility of betulin and its derivatives in various media is scattered and contradictory [50]. More in-depth research on the interaction of betulin with solvents has appeared only recently, and it is claimed that its solubility in pure organic solvents increases with increasing temperature.

Thus, it was found that the solubility of betulin in mixtures of acetone–water and ethanol–water increases with increasing temperature and the predominance of acetone or ethanol in the composition. The solubility of betulinic acid is 3 × 10^–3^ g/L [63], but it is not indicated for which polymorphic form it is determined. The water solubility lies in the range of 2.5 ± 0.5 g/L for most of the close triterpenoids. In the same work [63], the lipophilicity value Log Pow for betulin and lupeol is indicated, where Pow is the distribution coefficient between octanol and water, calculated in the ChemAxon program but not established experimentally. The lipophilicity value Log Pow is very large and is equal to 6.17 and 7.45 for betulin (**1**) and lupeol (**2**), respectively. The lipophilicity of triterpenoids characterized by Pow under the assumption that octanol has a lipophilicity similar to cell membranes is used as one of the indicators of drug absorption during passive diffusion. Compounds with low solubility in water and high Log Pow will be distributed more slowly from the cell membrane to the extracellular fluid (transcellular pathway) [64].

Thus, there are difficulties in stabilizing the structure and problems with dissolution and, accordingly, bioavailability, as well as the creation of injectable dosage forms for betulin and its derivatives. These compounds are an exception to the “rule of five” [65] and are capable of interaction both due to hydrogen bonds and hydrophobic binding, which leads to the formation of dimers (Figure 6), dense crystal packing arrangements and polymorphism.

### 1.3. Techniques for Increasing the Solubility of Pentacyclic Triterpenoids

One of the significant hindrances to conducting complex physico-chemical studies and effectively demonstrating the biological activity of the studied compounds is their limited solubility. The solubility and bioavailability of hardly soluble pentacyclic triterpenoids can be improved by various colloidal chemical approaches and by introducing hydrophilic groups through chemical modification, as shown for betulin (**1**) and its derivatives.

#### 1.3.1. Increasing the Solubility of Pentacyclic Triterpenoids Using Colloidal Chemical Approaches

There are a number of colloidal chemical techniques that allow us to increase the solubility of betulin (**1**) and its derivatives:The nanostructured introduction of betulin and its derivatives into liposomes and other nanoscale compounds [66,67]. Vesicles or liposomes from aqueous dispersions containing lecithin (for example, phosphatidylcholine) and triterpenoid dissolved in DMSO or alcohol are the most studied [68]. Nanoparticles with some betulin derivatives are obtained from polymers, while the most promising is an easily degradable copolymer of lactic and glycolic acids (PLGA) [67].The micelle formation of betulin and its derivatives with high-molecular compounds in which the polymer acts as a delivery vector [69,70].The use of physical principles for betulin and its derivatives (ultrasonic exposure, mechanochemical activation, etc.) in the presence of polymers (polyvinylpyrrolidone, polyethylene glycol, arabinogalactan, etc.) [71,72], which finds a positive effect in the demonstration of biological activity.The formation of supramolecular systems (inclusion complexes) [73] due to the complexation of betulin and its derivatives with γ-cyclodextrin, glycyrrhizic acid and other compounds capable of forming inclusion complexes by hydrophobic binding [73,74]. An analysis of such inclusion complexes showed, firstly, that the “guest” molecule is located in the cavity between the outer lipophilic sides of two betulin molecules. Secondly, betulin molecules are interconnected by hydrogen bonds so that in the crystal structure of the complex, there are dimers in which betulin molecules are connected by hydrogen bonds of the “head to tail” type (Figure 6). Individual betulin (**1**) in the crystal probably exists in the form of hydrogen-bound dimers of the “head to head” type, which, based on the data of vibrational spectra and X-ray diffraction analysis, is characterized for organic molecules [75]. Complexes including betulin and its derivatives with cyclodextrins in water dissociate into cyclodextrin and the active substrate, showing the biological properties of the latter [75,76,77,78].

The authors [73] showed the formation of host–guest complexes of betulin and its derivatives (allobetulin (**8**), betulinic acid (**5**), 3-ketobetulinic acid (**7**) with pyrazoles). Previously, complexes of betulin diacetate with metamizole sodium (1-phenyl-2,3-dimer-4-methylaminopyrazolone-5-*N*-methanesulfonate sodium) of the composition 1:1 were obtained. It can be assumed that with an optimal choice of reagents, it is possible to synthesize host–guest complexes with betulin (**1**) having the necessary hydrophilic–lipophilic ratio.

Triterpenoids are grafted onto polymers to create water-soluble polymer constructs. According to this approach, a terpene-containing monomer is usually obtained at the first stage, which is then introduced into radical copolymerization. For example, when betulin interacts with maleic acid in the presence of dicyclohexyl peroxidicarbonate at room temperature, the resulting monomer 28-*O*-betulin maleate copolymerizates with *N*-vinylpyrrolidone, acrylonitrile and vinyl acetate in the presence of radical initiators. The obtained polymers exhibit higher activity against RD TE32 rhabdomyosarcoma culture compared to betulin maleate [79].

Thus, it can be noted that increasing the bioavailability of betulin and its derivatives using colloidal chemical approaches, expressed in the production of various nanoparticles, both vesicular and micellar, is promising, although it does not solve the problem of sufficient solubility of lupan-type triterpenoids.

#### 1.3.2. Increasing the Solubility of Pentacyclic Triterpenoids by Chemical Modification of Betulin

Improving not only the bioavailability but also the solubility of triterpenoids, as well as any hydrophobic medicinal substances, can be achieved by chemical transformations by introducing various functional groups, such as carboxyl, hydroxyl, amino group, phosphate/phosphonate, carbonyl, etc. The chemical modification of the initial compounds—medicinal substances—includes the formation of esters, carbonates, carbamates, amides, phosphates, oximes and other derivatives, which are often considered prodrugs. The majority of betulin (**1**) chemical transformation methods are aimed at the functionalization of alcohol groups, the isopropenyl group and lateral cyclohexane and cyclopentane rings (Figure 1):

The most common method of chemical transformation of betulin (**1**) is a variety of reactions by alcohol groups (Figure 2).

As can be seen from Figure 2, the transformation processes of alcohol groups at the initial stage are aimed at esterification and oxidation reactions, which open the way to further functionalization with the formation of a wide variety of compounds.

The reactions of betulin in cycles **A** and **E** are mainly represented by transformations of lateral cyclohexane and cyclopentane rings through the processes of oxidation, halogenation, alkylation, condensation and intramolecular rearrangement (Figure 3).

The methods of chemical modification for the isopropenyl group of betulin are determined by the processes occurring with the participation of the olefin fragment and the functionalization of the terminal methyl group (Figure 4). One of the interesting and promising pathways of the functionalization of the isopropenyl group is the path to the creation of azacycles through the processes of aminolysis of intermediates.

Many studies have been devoted to the methods of chemical modification of betulin derivatives with biological activity, including in reviews [8,80,81,82,83]. Research on the targeted synthesis of betulin and betulinic acid derivatives with improved solubility began in 2000–2012, including the Gzuka review of patents [84].

Thus, summing up this part of the review, it should be noted that:Betulin and its derivatives are important strategic objects for medicine and pharmacy as drugs of anti-HIV, antitumor, hepatoprotective, hypolipidemic and other types of action.A necessary part of the research of betulin and its derivatives as potential medicinal substances is the study of polymorphism and the existence of structural modifications with various biopharmaceutical characteristics.A special place among the methods for obtaining betulin derivatives is occupied by modification methods to give hydrophilicity and, accordingly, better solubility and bioavailability, which in turn makes it possible to prepare hydrophilic dosage forms, including injection type.

## 2. Methods of Analysis of Betulin and Its Structural Analogues

As is known, plant species from different geographical locations can show a variable composition of total isolation products, which is also observed when extracting triterpenes from birch bark. The identification of birch bark in close connection with the geographical location is crucial to ensuring the quality and effectiveness of the raw material before it is converted into the target product. Considering that traditional approaches in analytical chemistry do not allow the rapid and reliable analysis of extractive substances, natural product manufacturers are constantly in search of faster and cheaper methods for checking their qualitative and quantitative composition, but most developed methods for the purification of betulin and its derivatives cannot achieve the criteria for the purity of the drug. Especially high purity requirements for betulin and its derivatives are imposed for dietary supplements, biochemical tests and medical practices. 

### 2.1. Identification of Birch Bark Species

A chemical indicator is used to identify the outer bark of fluffy and silver birch [85]. Thus, the diarylheptanoid glycoside platyphylloside, present in high concentrations in the inner bark of silver birch (20–30 mg/g) and in low concentrations in the inner bark of fluffy birch (<0.5 mg/g), makes it possible to distinguish these species using an indicator due to the rapid formation of orange sediment with platyphylloside.

### 2.2. Determination of Monosaccharides in Extracts for Pretreatment

Monosaccharides (in terms of hexoses) in birch bark extracts after pretreatment with hot water or an alkaline solution were determined by the Malaprade reaction. The 1,2-diols and 1,2,3-triols were oxidized by potassium periodate according to the method described in [86].

### 2.3. IR and Raman Spectroscopy

IR spectroscopy is the most effective and non-destructive analytical approach for the reliable identification of triterpenes obtained by various extraction methods, among the many instrumental methods of analysis [87,88,89,90]. For example, the ratio of the absorption band intensities at 1633 cm^−1^ is proposed to be used as a reference for comparing the content of triterpenes in birch bark [28,87,91,92]. Thus, ATR-FT-IR spectra of three birch bark extraction products (Romania) [89] were compared with those reported for isolated terpene compounds from *Betula utilis* and *Hyptis suaveolens* from the Himalayas [88] or *Betulaceae cortex* and *Betula pendula* Roth. [88] native to Central Europe. An analysis of IR spectra (KBr tablets) of Betulaceae extracts isolated by column chromatography in [88] led to the conclusion that triterpene substances, namely betulin, betulinic acid and lupeol obtained from birch bark (*Betula pendula* Roth.), give identical characteristic signals and optical density. However, comparison of the IR data of reference samples does not allow for distinguishing between betulin and lupeol. A detailed study conducted in another work [89] led to the conclusion that actual betulin demonstrates absorption bands in the ranges of 3430, 1716, 1641, 1600, 1581, 1291 and 881 cm^−1^, whereas betulinic acid has them at 3508, 1710, 1690, 1641, 1600, 1580 and 1290 cm^−1^, despite their high structural similarity.

In [93], vibrational FT-IR and FT-Raman spectroscopy were used as a fast, sensitive and effective method for the qualitative and semi-quantitative analysis of pentacyclic natural products of terpene nature obtained from the bark of Betula pendula Roth. growing in the Apuseni Mountains (Romania). A GC-MS analysis of extracts revealed two main compounds: betulin (59%) and lupeol (41%). The main IR/Raman spectra are presented for the isolated terpenes. The spectral imprint associated with pure triterpenes, where betulin bands predominated. It was found that betulinic acid is less noticeable as a separate compound during isolation, on the one hand because of its low concentration and, on the other hand, because of the weak Raman intensity characteristic for the –COOH group at 1681 and 1718 cm^−1^. Based on the analysis of IR and Raman spectra, the authors concluded that all extracts, regardless of the solvents used, revealed the dominant content of betulin and then lupeol. The work shows that Raman spectroscopy can provide semi-quantitative information regarding the content of triterpene in the final product by analyzing the ratio of the relative intensity of the 1642 cm^−1^ range (mode C=C) and 1601 cm^−1^, relative to other residual organic compounds from the bark. Since Raman measurements can also be performed on fresh plant material, this paper demonstrates the possibility of applying the results obtained to predict the highest triterpene content in tree bark for optimal collection time or for selecting individual genotypes directly in the field with the appropriate portable Raman equipment.

### 2.4. Chromatographic and Other Methods

The gas chromatographic separation of betulin-containing analytes is performed at elevated temperatures (above 300 °C) or after preliminary derivatization due to their low volatility [94], which makes the analysis long. Liquid chromatography by reverse-phase HPLC with spectrophotometric detection has found the widest application among the options for performing chromatographic analyses of betulin and its derivatives [48,95,96]. The absence of chromaphore groups that intensively absorb near-UV spectral regions in triterpenoid molecules determines their detection at wavelengths of 205–210 nm. Gradient elution worsens the reproducibility and reliability of the results of their analysis. In addition, this method is characterized by low sensitivity: the content limits are at the level of 0.1–1 mg/L.

The determination of betulin and betulinic acid in birch bark using RP-HPLC [48] extraction was tested using various solvents. The authors concluded that 95% ethanol in water is a good extraction solvent that allows the extraction of the high content of triterpenes. Separation was achieved in the reverse phase of the C18 column with acetonitrile/water 86:14 (vol./vol.). The detection was carried out using UV detection at λ = 210 nm. It was found that the content of triterpenes on the example of betulin and betulinic acid in the bark of white birch varies significantly depending on the place of growth in China [48].

High-performance liquid chromatography with reverse-phase (RP-HPLC) [9,15,48,97,98,99,100,101] and gas chromatography with mass spectrometric detection (GC-MS) [102,103] are widely used methods for analyzing samples of betulin and other triterpenes. Reverse-phase high-performance liquid chromatography (RP-HPLC) was used for the simultaneous extraction and determination of betulin and betulinic acid from white birch bark [48]. Betulin was isolated from the extract of dried carpophore Xanthoceras sorbifolia Bunge by chromatographic methods and identified based on spectral data [48].

Nevertheless, thin-layer chromatography (TLC, especially in the high-performance modification—HPTLC) is also well applicable for assessing purity and visualizing impurities [88,104,105].

In [106], one of the fractions of betulin extract was crystallized from an ethanol solution in the form of flat transparent colorless small crystals with an average size of about 0.2 mm in length. Much larger crystals of up to 12 mm long are formed if a hot saturated ethanol solution is slowly cooled to room temperature. Betulin crystallizes from an ethanol solution with orthorhombic symmetry with one ethanol molecule per one hydrogen-bonded betulin molecule [14,51,107]. One of the classical physico-chemical characteristics of pure substances is their melting point. In the literature, there is a wide range of melting temperatures attributed to betulin 251–261 °C [51]. Most authors are inclined to a value of about 255 °C. Thus, when the betulin sample was heated, the following crystal changes were observed: 130–140 °C—darkening of crystals due to ethanol evaporation; 160–170 °C—betulin began to sublimate noticeably; 180–200 °C—new transparent microcrystals appeared on the surface of the initial crystals; 255–256 °C—total melting. Crystals did not form after cooling, and the melt solidified in the amorphous glassy form. Amorphous microscopic beads were also formed in the form of a bright white powder during sublimation at atmospheric pressure in a sublimator [106].

Birch bark consists of ~75% brown inner bark and ~25% white outer bark. The outer bark contains fats, fatty acids, resins, suberin and especially betulin—up to 30% [6]. In [106], a simple approach was used to microscopically determine the location of triterpenes in birch bark. The authors made a lateral section of the bark to which Nile red dissolved in dimethyl phthalate was added. Fluorescence microscopy was used to obtain the image. Nile red fluorescent dye is very lipophilic, and, therefore, is used for labeling lipophilic structures. Thus, this method allows the identification of brighter areas in the photo that indicate the location of hydrophobic triterpenes. Betulin is especially clearly determined when using fluorescence microscopy. The structure of the birch bark, revealed after marking with Nile red, allowed us to establish that betulin is localized by small longitudinal shells passing through the birch bark. Taking into account the presence of triterpene shells and the heterogeneity of the localization of betulin particles, the authors point out the need to pay due attention to the mechanical processing of birch bark before the extraction process. This observation is confirmed by the fact that the bark cut into small pieces (about 5 × 5 mm) gave, on average, only about 19% of extractives, whereas grinding to 4 µm gave more than 45% of extractives. 

The structures of betulin and its derivatives as well as the qualitative and quantitative compositions of dry and liquid extracts were established using a complex of physico-chemical analysis methods: TLC, IR spectroscopy, ^1^H and ^13^C NMR spectroscopy and chromatography-mass spectrometry. Thus, the identification of the betulin conversion product—betulin diacetate—was carried out by NMR spectroscopy in [108]. The authors of [109] studied the composition of birch bark extract, while the quantitative content of betulin was determined using the chromatography-mass spectrometry method. 

An obvious alternative to the spectrophotometric detection of triterpenoids in liquid is mass spectrometry, which is characterized by high sensitivity, extremely high selectivity and a wide dynamic range. The use of HPLC mass spectrometry for the determination of lupane, oleanane and ursan and their comparison with other methods is shown in [110], and it is established that the best results can be obtained by chemical ionization at atmospheric pressure (APCI) and photochemical ionization at atmospheric pressure (APPI) using toluene as a dopant. The achieved detection limits for various compounds were in the range of 5–15 mcg/L and 2–840 mcg/L for APCI and APPI, respectively. It was shown that in the case of chemical ionization, high sensitivity was achieved for betulinic acids, and for other triterpenoids, the detection limits were at the level of hundreds of mcg/L. The method of increasing the sensitivity and reliability of the determination of triterpenoids, especially in the analysis of complex matrices using tandem mass spectrometry, was successfully used in [111] for blood plasma analysis in the study of the pharmacokinetics of betulonic acid with a detection limit of 3 mcg/L. 

The ion-induced dissociation (IID) spectra of betulonic acid derivatives along with the spectra of other triterpenoids containing carboxyl groups were studied in [112] for the reliable identification of analytes. A wider range of triterpenoids was investigated in [113]. The similarity of the spectra of compounds carrying saturated cycles with the spectra of electron ionization with a predominance of ring rupture was demonstrated [113]. It has been observed that triterpenoids with an unsaturated skeleton (olean and ursan) undergo fragmentation by a more complex mechanism due to the possibility of double bond migration. In the development of these works [114], the HPLC MS/MS method of identifying and determining the four main components of birch bark extracts (betulin, lupeol, betulinic acid and erythrodiol) was developed (Figure 7).

The developed method of the highly sensitive and selective determination of betulin, lupeol, erythrodiol and betulinic acid is proposed based on a combination of chromatographic separation with tandem detection of mass spectrometry. It was found that the APCI/MS/MS method is preferable and provides detection limits of the studied compounds at the level of several parts per billion and is characterized by a linear dynamic range spanning more than three orders of magnitude.

It has been shown that the insufficient purity of substances isolated from extracts or reaction mass containing target substances is a serious hindrance for voltametric studies [115] during the determination of antioxidant activity as well as the further chemical modifications of triterpenes. Obviously, such studies dictate the need for simple ways to clean them from accompanying impurities. Therefore, one of the urgent tasks is to develop simple and effective ways to purify substances containing triterpenes. We have already noted above that currently, the literature information on the methods of purifying triterpenes is presented, as a rule, by methods of recrystallizing betulin and its derivatives from various solvents (such as lower aliphatic alcohols, chloroform, acetone, ethyl acetate, dichloromethane, etc.). However, these techniques are associated with the formation of solvate complexes with triterpenoids, which makes it difficult to identify them as individual compounds [62]. In addition, significant losses (up to 40%) of the target substances are observed when using ethanol as a solvent for recrystallization [62]. To overcome such hindrances, the methods of purification of representatives of the lupane and oleane series using recrystallization and column chromatography methods were considered in [116], and the effectiveness of these methods was compared. The best result was achieved when dry extracts were purified by column chromatography with aluminum oxide as a sorbent and acetone as an eluent. In this case, according to chromato-mass spectrometric analysis, an increase in the content of the main substance in the sample by 4–7% to 95–98% levels was recorded.

It was shown that this method is convenient for the purification of betulin, betulin diacetate and allobetulin from concomitant impurities and allows us to obtain their pure substances suitable for further use in organic synthesis, electrochemical, microbiological and other research methods. The advantages of the proposed chromatographic method of purification of triterpenoids are the simplicity of execution, expressiveness and high purity of the obtained samples. 

A systematic spectral and chromatographic study was carried out for the isolated pentacyclic triterpenoids (betulin (**1**), betulin diacetate (**9**) and allobetulin (**8**)) using a number of physico-chemical analysis methods (IR, PMR, GCMS) in [117]. The conditions were selected to ensure the suitability of TLC analysis for the qualitative detection of the listed compounds, and conditions of HPLC analysis for qualitative and quantitative determination of betulin were found and described.

The authors systematized the obtained spectral (IR, NMR, GCMS) and chromatographic (TLC, HPLC) data on the physicochemical properties of the most important representatives of the lupane and oleane series—betulin (**1**), betulin diacetate (**9**) and allobetulin (**8**), the structures of which are shown in Figure 8.

### 2.5. The Analysis of 3β,28-Dihydroxy-20(29)-lupen (1), Betulin (**1**)

In the IR spectrum of betulin (**1**), there is a wide absorption band at 3359 cm^−1^ belonging to valence vibrations of hydroxyl groups (Table 1). The stretching modes of the =CH_2_ group are observed at 3078 cm^−1^; the ones of the C–H groups are observed as a series of bands at 2944 and 2870 cm^–1^. The bending vibrations of a multiple C=C bond are detected at 1695 and 1644 cm^−1^. The vibrations of the CH_2_ groups are attributed to 1457 cm^−1^, and those of the CH_3_ group are attributed to the band at 1374 cm^−1^. The stretching mode of the C–O group is observed as a strong band at 1029 cm^−1^.

In the ^1^H NMR spectrum of betulin (**1**), there are two singlet signals in the region of 4.58 and 4.68 ppm, referring to the chemical shift of =CH_2_ multiple bond protons (Table 1). Also, the spectrum is characterized by the presence of two doublet signals at 3.32 and 3.79 ppm, referring to protons at the 28th position. The proton in the 3-position is marked by the presence of a quadrupled signal at 3.18 ppm. The complex signal corresponded to a proton at the 19th carbon atom appearing at 2.38 ppm. There is an ensemble of signals of the CH_3_, CH_2_ and CH groups in the region of 0.65–2.15 ppm.

In the mass spectrum of betulin (**1**), there were peaks of high fragment intensities of 203, 189, 135, 121, and 95 (M^+^), characteristic of the lupan skeleton along with a small peak of the molecular ion 442 (M^+^) (Table 1).

### 2.6. The Analysis of 3β,28-Diacetoxy-lup-20(29)-en(9), Betulin Diacetate (**9**)

The stretching modes of =CH_2_ groups were located at 3066 cm^−1^, and the modes of alkyl fragments of C–H groups were observed as a series of bands at 2945 and 2869 cm^−1^ in the IR spectrum of betulin diacetate 2 (Table 1). The vibrations of the carbonyl C=O group were detected as a strong absorption band at 1733 cm^−1^. The bending vibrations of the CH_2_ and CH_3_ groups were attributed to 1457 cm^−1^ 1367 cm^–1^, respectively. The strong band at 1241 cm^−1^ corresponded to the stretching mode of the C–O group.

In the ^1^H NMR spectrum of betulin diacetate (**2**), there were two singlet signals in the region of 4.58 and 4.68 ppm, referring to the chemical shift of =CH_2_ multiple bond protons (Table 1). There was a quadruplet signal at 4.46 ppm, corresponding to a proton in the 3-position. Two doublet signals at 3.84 and 4.24 ppm were assigned to proton bonds at the 28-position. The signal at 2.43 ppm referred to a proton at the 19 carbon atom. There was a complex of signals of the CH_3_, CH_2_ and CH groups in the region of 0.74–2.15 ppm.

In the mass spectrum of betulin diacetate (**2**), along with an insignificant peak in the intensity of the molecular ion 526 (M^+^), there were peaks of high intensity of fragments of 189, 135, 121 and 91, characteristic of the lupan backbone, as well as peaks of medium intensity of 466, 423, 203, 161, 67 and 55 (Table 1).

### 2.7. The Analysis of 3β-Hydroxy-19β,28-epoxy-18α-olean (**8**), Allobetulin

In the IR spectrum of allobetulin (**3**), there was a broadened absorption band at 3430 cm^−1^ (Table 1) assigned to the stretching modes of the hydroxyl group. The stretching modes of C–H groups were observed as a series of bands at 2939 and 2867 cm^−1^. The bending vibrations of CH_2_ and CH_3_ groups were attributed to a band at 1450 cm^−1^ and a series of bands in the regions of 1386, 1375 and 1361 cm^−1^, respectively. The strong band at 1041 cm^–1^ corresponded to the stretching mode of the C–O group.

In the ^1^H NMR spectrum of allobetulin (**8**), there were two doublet signals at 3.77 and 3.44 ppm, related to protons –CH_2_ bonds at the 28-position of the carbon atom (Table 1). Also, in the spectrum, there was a singlet signal at 3.52 ppm corresponding to the proton of the CH group at the 19th position. The multiplet signal in the region of 3.19 ppm was assigned to the proton of the CH group at the 3rd carbon atom. There was an ensemble of signals of the CH_3_, CH_2_ and CH groups in the region of 0.65–1.82 ppm.

In the mass spectrum of allobetulin (**8**), along with the peak of the molecular ion 442 (M^+^), there were peaks of high fragment intensity of 207, 189, 134, 107, 95, 81, 69 and 55, characteristic of the olean backbone, and there were peaks of a medium intensity of 177, 148 and 121 (Table 1).

In the second stage of this work, the chromatographic characteristics (TLC, HPLC) of the dry substances of betulin (**1**), betulin diacetate (**9**) and allobetulin (**8**) were studied.

The most effective eluting systems (A–C: ethyl acetate: hexane = 1:1 (A), hexane: ethyl acetate = 4:1 (B), chloroform: ethanol = 40:1 (C) were selected to obtain correct TLC analysis data for the qualitative determination of compounds **1**, **8** and **9**. A sample of betulin (**1**) was chromatographed in system A; the Rf index was 0.40. A sample of betulin diacetate (**9**) was chromatographed in system B (Rf 0.66), and a sample of allobetulin (**8**) in system C (Rf 0.37). The appearance of spots in all cases was carried out with a modified Ehrlich reagent since no spots are detected when the plates are UV irradiated with photons of a standard wavelength of 254 nm.

The betulin (1) samples were examined by HPLC. A sample of betulin (**1**) of a high degree of purification (with a content of 98% according to HGMS data) was used as a working standard (WS). The test samples of botulin and WS and the placebo samples were dissolved in acetonitrile with a concentration of 1 mg/mL. The samples were centrifuged for 10 min to remove the insoluble particles and placed in an autoclave and chromatographed using an Agilent 1260 Infinity liquid chromatograph with a diode-matrix detector. The column measuring 100 × 4.6 mm (inner diameter) was made of stainless steel and equipped with a pre-column. Zorbax Extend-C18 Rapid Resolution 3.5 microns 80 Å was used as the stationary phase. The mobile phase was a mixture of solvents: water—acetonitrile = 3:7. Isocratic elution was carried out. The temperature of the column was 40.0°C. The volumetric flow rate was 0.6 mL/min. The analysis time was 30 min. The temperature of the detector cell was 40.0 °C. The substances were detected using four wavelengths: 200 nm, 220 nm, 230 nm and 250 nm.

The selected conditions ensured the suitability of the technique for the qualitative and quantitative analysis of betulin (**1**). At the same time, the retention time for betulin (**1**) corresponded to 17.2 min (Figure 9). The corrected retention time was 15.5 min. The retention factor was at least 9. The efficiency of the betulin column was at least 50,000 theoretical plates.

The currently available methods [62] of purifying betulin are mainly based on its recrystallization from polar protons and aprotic solvents. As a rule, the use of solvents for recrystallization, such as lower aliphatic alcohols, chloroform, acetone, ethyl acetate and dichloromethane, leads to the formation of solvate complexes with betulin. It should be noted that these solvents have low selectivity to concomitant impurities of betulin, such as lupeol, polyphenols and oxidized forms of betulin (betulin and betulonic acids and aldehydes). In addition, the use of these solvents for recrystallization leads to significant losses of betulin, which can reach 40%.

The possibility of purifying betulin (**1**) using a high-boiling hydrocarbon solvent was investigated to achieve a higher degree of purity. Nonane, which has a boiling point of 151 °C, was taken as an example for research and repeatedly returning into the presence of betulin until it was completely dissolved [118,119]. It has been established that the use of nonane as a solvent for recrystallization of betulin (**1**) makes it possible to obtain betulin with a purity of up to 98%.

It is important to note that the use of a high-boiling hydrocarbon solvent, such as nonane, avoids the formation of stable betulin solvate complexes and ensures the selective isolation of compounds included in the total alcohol extract of birch bark due to differences in their physico-chemical properties

Thus, the proposed method makes it possible to effectively separate valuable products (such as betulin), polyphenol fractions and phytosterols contained in extractive plant materials using a simpler and cheaper technological scheme. The proposed method makes it possible to obtain betulin with a purity of 98%, suitable for use in subsequent chemical transformations without additional purification. In addition, a fraction of substances that are valuable for further use and contain polyphenols and phytosterols was obtained as a by-product. During the development of betulin purification technology, it was shown that the used solvent (nonan) could be reused.

This article is a logical continuation of the work [120].

## 3. Conclusions

Summarizing the generalized material on the methods of analysis and identification of betulin and some of its related compounds, it can be concluded that the analytical control of their quality is crucial for obtaining targeted pentacyclic triterpenoids with specified properties, especially for testing biological activity in food supplements and cosmetics.

Since betulin oxidation products are precursors of many biologically active compounds—potential drugs—in our opinion, generalized physicochemical characteristics of these compounds will be extremely useful to researchers in this field. Below, we have summarized the literature data on IR and NMR spectroscopy in the table (see Appendix A) and given the melting temperatures of key acids and aldehydes based on betulin.

## Data Availability

Not applicable.

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
