# Peer review of "Methods of Analysis and Identification of Betulin and Its Derivatives"

_molecules, 2023, doi:10.3390/molecules28165946_

Round 1
Reviewer 1 Report
1. The text contains large fragments in Russian language (especially the list of references). In addition, the caption for figure 4;
2. Reference numbering should be checked and corrected (for example, ref. 51 is missing);
3. The numbering of sections in the Introduction is not correct too;
4. The quality of schemes 2-4 needs to be improved;
5. it is not clear why the reference to the table with the literature data of IR and NMR spectra appears only in the Conclusion?
Author Response
Dear Reviewer.
1 The list of references has been completely corrected. Figure 4 corrected inscription. Now it is in English.
2 Fixed link numbering.
3 The numbering of sections has been corrected.
4 Improved the quality of schemes 2-4.
5 the link to appendix 1 is indicated in the conclusion and in the updated article I inserted the appendix itself directly.
Reviewer 2 Report
1. The research work of the article is not reflected in the abstract;
2. The innovation and application value of this study are not prominent enough, but simply state the relevant processes and strategies;
3. The structure of the article is quite scattered, and the numbering of some subheadings is also messy;
4. Line 176-178,Is there any data support, such as the results of theoretical calculations; Line 609-610, Is there any evidence for the rearrangement of allobetulin 8 into betulin 1;
5. There are still some issues in the text, such as: the caption text in Figure 4 & table1 is not in English, and 3066 (C=C) in table 1 is incorrect.
1. The research work of the article is not reflected in the abstract;
2. The innovation and application value of this study are not prominent enough, but simply state the relevant processes and strategies;
3. The structure of the article is quite scattered, and the numbering of some subheadings is also messy;
4. Line 176-178,Is there any data support, such as the results of theoretical calculations; Line 609-610, Is there any evidence for the rearrangement of allobetulin 8 into betulin 1;
5. There are still some issues in the text, such as: the caption text in Figure 4 & table1 is not in English, and 3066 (C=C) in table 1 is incorrect.
Author Response
This scientific work presents practical and theoretical material on the methods of analysis and identification of betulin and its key derivatives.
The properties of betulin and its derivatives, which are determined by the structural features of this class of compounds and their tendency to form dimers, polymorphism, and isomerization, are considered.
The article outlines ways to improve not only the bioavailability, but also the solubility of triterpenoids, as well as any hydrophobic drug substances, through chemical transformations by introducing various functional groups, such as carboxyl, hydroxyl, amino, phosphate/phosphonate and carbonyl.
The authors of the article summarized the physicochemical characteristics of betulin and its compounds, systematized the literature data on IR and NMR spectroscopy and gave the melting temperatures of key acids and aldehydes based on betulin
Round 2
Reviewer 2 Report
the abstract should be revised,and the cited references too.